# Long-Read Whole-Genome Sequencing as a Tool for Variant Detection in Inherited Retinal Dystrophies

**DOI:** 10.3390/ijms26083825

**Published:** 2025-04-18

**Authors:** Cristina Rodilla, Gonzalo Núñez-Moreno, Yolanda Benitez, Marta Rodríguez de Alba, Fiona Blanco-Kelly, Aroa López-Alcojor, Lidia Fernández-Caballero, Irene Perea-Romero, Marta Del Pozo-Valero, Gema García-García, Mar Balanzá, Cristina Villaverde, Olga Zurita, Claire Jubin, Cedric Fund, Marc Delepine, Aurelie Leduc, Jean-François Deleuze, José M. Millán, Pablo Minguez, Marta Corton, Carmen Ayuso

**Affiliations:** 1Department of Genetics & Genomics, Instituto de Investigación Sanitaria-Fundación Jiménez Díaz University Hospital, Universidad Autónoma de Madrid (IIS-FJD, UAM), 28040 Madrid, Spain; cristina.rodilla@quironsalud.es (C.R.); gonzalo.n.moreno@gmail.com (G.N.-M.); yolanda.bquesada@quironsalud.es (Y.B.); mrodrigueza@fjd.es (M.R.d.A.); fblancok@quironsalud.es (F.B.-K.); arlo.edu@quironsalud.es (A.L.-A.); lidia.fernandezc@quironsalud.es (L.F.-C.); ireneperearomero@gmail.com (I.P.-R.); martapova21@gmail.com (M.D.P.-V.); cvillaverde@fjd.es (C.V.); olga.zurita@fjd.es (O.Z.); pablo.minguez@quironsalud.es (P.M.); 2Center for Biomedical Network Research on Rare Diseases (CIBERER), Instituto de Salud Carlos III, 28029 Madrid, Spain; gema_garcia@iislafe.es (G.G.-G.); mar_balanza@iislafe.es (M.B.); jose_millan@iislafe.es (J.M.M.); 3Bioinformatics Unit, Instituto de Investigación Sanitaria-Fundación Jiménez Díaz University Hospital, Universidad Autónoma de Madrid (IIS-FJD, UAM), 28040 Madrid, Spain; 4Molecular, Cellular and Genomics Biomedicine, Health Research Institute La Fe, 46026 Valencia, Spain; 5Centre National de Recherche en Génomique Humaine, Université Paris-Saclay, 91057 Evry, France; cjubin@cnrgh.fr (C.J.); cedric.fund@cnrgh.fr (C.F.); marc.delepine@cnrgh.fr (M.D.); aurelie.leduc@cnrgh.fr (A.L.); deleuze@cng.fr (J.-F.D.)

**Keywords:** long-read sequencing, nanopore sequencing, inherited retinal dystrophies

## Abstract

Advances in whole-genome sequencing (WGS) have significantly enhanced our ability to detect genomic variants underlying inherited diseases. In this study, we performed long-read WGS on 24 patients with inherited retinal dystrophies (IRDs) to validate the utility of nanopore sequencing in detecting genomic variations. We confirmed the presence of all previously detected variants and demonstrated that this approach allows for the precise refinement of structural variants (SVs). Furthermore, we could perform genotype phasing by sequencing only the probands, confirming that the variants were inherited in trans. Moreover, nanopore sequencing enables the detection of complex variants, such as transposon insertions and structural rearrangements. This comprehensive assessment illustrates the power of long-read sequencing in capturing diverse forms of genomic variation and in improving diagnostic accuracy in IRDs.

## 1. Introduction

The rapid evolution of DNA sequencing technologies has profoundly expanded our ability to explore genetic variation, providing insights into the molecular mechanisms that drive the onset of human diseases. Inherited retinal dystrophies (IRDs) constitute a group of rare clinically and genetically heterogeneous diseases that primarily affect photoreceptor and retinal pigment epithelium (RPE) cells. For many years, short-read sequencing platforms, such as Illumina, have been the gold standard for the molecular diagnosis of IRD patients, providing high accuracy and cost-efficient studies [1,2]. However, despite their wide adoption, short-read technologies are inherently limited by their read length.

Whole-genome sequencing (WGS) has revolutionized the field by enabling the identification of single nucleotide variants (SNVs), insertions/deletions (indels), copy number variants (CNVs), and other forms of genetic variation, especially those lying in non-sequencing regions of the genome that may be missed by more targeted approaches, such as exome sequencing [3,4]. By capturing both coding and non-coding regions of the genome, WGS allows for a more complete understanding of the genetic basis of diseases, particularly in cases involving non-coding regulatory elements or structural rearrangements that affect gene function. However, WGS performed with short-read sequencing still faces critical challenges, especially in regions of the genome that encompass structural variants (SVs), long tandem repeats, or GC-rich sequences, all of which are difficult to resolve with short reads.

Traditionally, cytogenomic studies have been alternative approaches to SV detection and offer a genomic resolution limited to 5–10 Mb for karyotyping, >100 kb for fluorescence in situ hybridization (FISH), and 20–200 kb for array comparative genome hybridization (aCGH), although balanced SVs are undetectable with the latter [5].

To address these limitations, long-read sequencing technologies, particularly nanopore sequencing, have emerged as essential tools for comprehensive genome analysis [6]. Nanopore sequencing offers the ability to produce reads exceeding tens of kilobases in length, providing a more contiguous and detailed view of the genome. This makes it possible to directly sequence through complex regions, identify large structural variants, and accurately phase alleles across long stretches of DNA [7,8]. Importantly, nanopore sequencing can be applied to whole-genome sequencing, further enhancing its utility in resolving variants across the entire genome, including previously inaccessible regions.

Moreover, nanopore sequencing allows for the direct sequencing of native DNA and RNA molecules without the need for amplification, thereby reducing biases introduced by PCR. This feature also facilitates the detection of epigenetic modifications, such as DNA methylation, which can play critical roles in gene regulation and disease pathogenesis [9]. Furthermore, nanopore sequencing can be performed in real time, offering immediate insights into data as they are generated, a feature that is particularly useful in time-sensitive diagnostic or epidemiological applications.

In this study, we aimed to evaluate the possible clinical applications of nanopore-based long-read WGS (LR-WGS) and its current limitations for the genetic diagnosis of IRD patients, underscoring its growing role in both research and clinical diagnostics.

## 2. Results

We performed LR-WGS on 24 IRD patients to assess the efficacy of nanopore sequencing, focusing on the detection of structural variation. The cohort comprised eleven syndromic retinitis pigmentosa (RP) patients, six patients with Stargardt disease, five RP patients, one retinal dystrophy patient, and one patient with best vitelliform macular dystrophy (Appendix A). More than half of the cohort (16 out of 24) reported no affected relatives. Almost all patients (23 out of 24) had previously undergone CES analysis, while the remaining patient had undergone a WES study. Five of these patients had been molecularly characterized by CES analysis and were selected as positive controls. A detailed description of patients’ initial molecular findings and newly identified variants within the studied cohort are described in Appendix A.

### 2.1. Confirmation of Previously Detected Variants in the Studied Cohort

There was a total of 24 different previously reported variants in the 24 probands included in this study (Figure 1A,B). These variants included nineteen SNVs, which could be classified according to their mutational effect into fifteen missense variants, two nonsense variants, one splicing variant, and one deep-intronic variant. There were also four reported CNVs, which included two deletions and two duplications. Finally, there was a single nucleotide insertion variant reported in the cohort that encoded for a frameshift variant.

All the 24 previously reported variants in the studied cases were confirmed using nanopore sequencing.

Exome sequencing in index patients C002319 and C002311 identified variants in *ADGRV1* and *PROM1*, respectively. Segregation analysis revealed that the *ADGRV1* variants were in cis, forming a complex allele in the patient, while the *PROM1* variant was inherited from an unaffected mother, thereby excluding a dominant inheritance pattern. Additionally, two variants were found in other genes known to be associated with both autosomal dominant and recessive inheritance: *BEST1* and *IMPG2*. However, the available information from genomic databases supported a recessive inheritance for both. The *BEST1* variant affects the p.Pro101 residue, whereas other pathogenic variants (c.301C > A and c.302C > T) have been linked to autosomal recessive *BEST1*-related conditions (ClinVar IDs: 99710 and 877528). The *IMPG2* variant (NM_016247.4:c.1300C > T) is observed in general population databases at a frequency (GnomAD v4, f = 1.2 × 10^−4^) higher than expected for a dominant pathogenic variant, supporting a non-dominant model of inheritance.

LR-WGS also allowed us to define the exact structural variation that the characterized patients (validation dataset) carried.

A 38-year-old male who was clinically diagnosed with RP (C001VXB) presented with disease onset at 17 years of age. The patient reported a decreased visual field, night blindness, and photophobia. Two variants in the *EYS* gene had been previously detected via CES analysis. One pathogenic indel variant, NM_001142800:c.4103dupT, located in exon 26, and a heterozygous deletion of exons 14 to 22, were detected. The deletion was reported as a range NC_000006.12:g.(64626376_64813368)_(64997811_65057603)del and since the exact breakpoints were unknown, its size varied from 0.18 Mb–0.43 Mb. The deletion of exons within this variant was confirmed via MLPA study (Figure 2B).

LR-WGS confirmed that both variants were detected in this patient. Long-reads also allowed for the precise breakpoint definition of the previously detected deletion (chr6:64738649-65055624) (Figure 2C), giving the variant an exact size of 316,975 bp. This information defines the effect of the variant on the associated transcript NM_001142800.2:c.2137 + 1990_3443 + 74729del and reduces the protein from 3144 amino acids to a predicted 726-amino-acid sequence. The start position of the deletion is included within a repeat region of the hAT-Tip100 family, and the end position is located in a LINE1 element.

### 2.2. Limitations of the Technique

Although all previously detected variants were confirmed in the LR-WGS data, there were two variants that required further inspection, since the variant callers used were not able to report them.

The one-nucleotide indel of the C001VXB patient was detected in only some reads, and its alignment was heterogeneous (Figure 1C).

On the other hand, the duplication in chromosome 16 of patient C00231A previously identified with exome sequencing (CES and WES) and aCGH was not detected via the SV caller or manual exploration. This variant could be identified only with a coverage comparison between C00231A and the other studied patients (Figure 1D). For this reason, only the duplication interval is determined, and the exact breakpoints could not be determined.

### 2.3. Genotype Characterization in Recessive IRDs

A male with RP, hearing loss, and behavioral problems (C002312) had presented vision problems since infancy. CES analysis on the patient did not yield a conclusive molecular diagnosis, while WES identified a heterozygous truncating variant in the candidate gene *ARHGEF16*, which, although not associated with any disease, is ubiquitously expressed in human tissues and was considered a potential candidate for the patient’s syndromic phenotype. After LR-WGS, two variants were identified in a recently associated IRD gene (included in PanelApp in May 2024, Retinal disorders panel v7.0 October 2024), and previous NGS studies were performed before 2018. CFAP20 is a ciliar protein implicated in microtubule architecture [10], and has been reported to be the molecular cause of non-syndromic IRD in five independent families [10,11]. One detected variant, NM_013242.3:c.257A > G, is a missense change not reported in GnomAD, submitted as likely pathogenic in ClinVar, and inherited in trans in three affected RP siblings from the same family [10]. The other variant, NM_013242.3:c.277–207_*57060del, is a 58.7 kb deletion (chr16:58056967-58115666) that extends from intron 3 to the end of the gene.

Variants were confirmed via Sanger sequencing, with the deletion segregated from the unaffected father (Figure 3A). The origin of the SNV remains undetermined. However, long-read sequencing revealed that both variants were located in *trans* in the proband (Figure 3C).

A male with RP, congenital profound sensorineural hearing loss, and vestibular impairment, who was diagnosed with Usher syndrome type I (C00231G), remained non-informative and was included in the study (Figure 3A). Two variants were identified in the *MYO7A* gene in the LR-WGS data, which could explain his phenotype. The first variant, NM_000260.3:c.1657C > T, implies a missense change in exon 14. This variant is reported in GnomAD with an extremely low frequency (f = 6.2 × 10^−7^), and in silico predictors support its pathogenic effect. The second variant is a 16.9 kb deletion (NM_000260.3:c.5945-60_*10804del; chr11:77208637-77225500del) from intron 43 of *MYO7A* to the adjacent gene *GDPD4*. Both variants were confirmed via Sanger sequencing and segregated in the unaffected father and brother (Figure 4A). Since the distance between both variants was 45.7 kb, phasing was performed, and the resulting haplotypes confirmed that both variants were located in *trans* in the proband (Figure 4C).

### 2.4. Repeat Mobile Element Variation

A 43-year-old woman with a clinical diagnosis of Stargardt disease (C001VWE) had presented with reduced visual acuity since the age of six. Two calls for a centromeric inversion on chromosome 1 were detected in the LR-WGS data (Appendix A). However, karyotype analysis revealed no G band pattern alterations at the performed resolution level.

Ribbon visualization revealed the inverted intrachromosomal insertional translocation of a region located in the 1p31.1 cytoband to the 1q42.3 cytoband (Figure 5C). The region is 6068 bp in size; in this location (1p31.1) there is a reported transposon L1HS element of 6054 bp. The insertion site in the 1p42.3 cytoband is an intergenic region with low evolutionary conservation, and does not co-locate with any reported *cis*-regulatory elements, relevant topologically associated domains, or retinal transcription factor sites.

### 2.5. Non-Causative Complex Structural Rearrangement

In the female patient presented above (C001VWE), there were multiple SV calls outside the pseudoautosomal region in chromosome X, one of which implied a deletion of 84 Mb (Appendix A). Further exploration of the regions involved in IGV and Ribbon showed that there had been a duplication of different regions of the Xq23 cytoband that had been reorganized and inserted into the Xp21.3 cytoband, where other regions were duplicated as well (Figure 6).

aCGH confirmed a duplication in Xp21.3 [chrX:25293248-25666268] and Xq23 [chrX:109615463-110084362] with similar breakpoints in the proband and healthy father, thus validating the LR-WGS finding. The presence of the variant in the unaffected father supports a non-deleterious effect.

## 3. Discussion

The emergence of long-read sequencing technologies, particularly nanopore sequencing, has provided the ability to resolve genomic variants that are otherwise difficult to detect using traditional sequencing and cytogenomic approaches. Nanopore sequencing provides reads that can span tens of kilobases and even megabases [12,13], allowing for the more accurate assembly of repetitive regions, detection of large structural variants, and phasing of alleles.

In this study, we utilized nanopore long-read sequencing, not only to confirm previously detected second-generation NGS variation, but also to delineate more precisely the structural variant responsible for a patient’s condition by defining the exact breakpoints. The high-resolution mapping provided by nanopore sequencing enhances the ability to interpret variant effects that may otherwise remain ambiguous or undetected through traditional short-read methods [14], thereby increasing diagnostic accuracy and clinical relevance.

Through this long-read whole-genome sequencing (LR-WGS) study, we were able to molecularly characterize two additional patients out of nineteen previously unsolved index cases, representing a diagnostic yield of 10.5%. This result is comparable to that reported by Negi et al., who achieved an 11.2% diagnostic rate (11/98) in patients with suspected rare monogenic disorders that had remained unsolved after short-read NGS [15]. Similarly, another study focusing on monoallelic *EYS* cases previously analyzed by NGS identified a genetic diagnosis in 13.3% of the cohort (2/15) using long-read WGS [16].

We have also resolved the inheritance pattern in these two patients, which otherwise could not have been determined using other sequencing methods. The C002312 patient was determined via the direct visualization of the obtained reads in the region and the C00231G patient by performing proband-only haplotype assembly. Therefore, the reliance on DNA samples from additional family members, which was until recently necessary for variant phasing or determining haplotypes, is greatly reduced with the use of long-read sequencing on the proband [15,17,18]. This not only simplifies the steps, but also accelerates the process of reaching a molecular diagnosis.

We further confirmed the ability of nanopore sequencing to detect repeat mobile element activity in the genome, identifying a LINE-1 retrotransposon inverted insertion in chromosome 1. LINE-1s are ∼6 kb in size and are the only known autonomously active human retrotransposons. Although transposon movement has already been associated with IRD causing variation [19,20], polymorphic transposon variation is still well under-detected and under-reported [21].

This last point also applies to complex structural rearrangement detection. We presented a complex chromosomal variation in chromosome X detected in a patient, that could only be precisely detected using long-read sequences. Manual variant interpretation after SV calling was crucial for correctly defining the variant configuration. Segregation in unaffected parents confirmed it had been inherited from her father. After this finding, the variant could be classified as likely benign because it was carried by an unaffected male in a hemizygous pattern. However, the lack of structural variation databases for the general population hinders clinical significance associations [6,15,22,23,24].

Despite its many advantages, nanopore sequencing has some limitations. Although the detection of point genetic variants has improved, the error rate is still higher than that obtained from short-read technologies such as Illumina [25] or other long-read platforms, such as PacBio [26]. In this study, one previously detected small indel was not called. Indel detection in nanopore data is still a challenge, especially in homopolymer regions [9,21]. Reads containing an indel in this location were not enough for it to be considered heterozygous. Additionally, if a consensus position is not reached, the variant caller may be unable to determine its presence.

Furthermore, we have also detected errors in structural variant detection. Deletions were precisely identified, with a size range from 6.3 kb up to 317 kb and duplications from 110 pb (i.e., the one contained within the complex rearrangement located in the X chromosome of the C001VWE patient) to 611 kb. However, the exact breakpoints of the 16p13.11 duplication of the C00231A patient could not be defined using this technology. This cytoband has been previously reported to be prompt in low copy repeat-mediated duplications differing in size, with the specific one reported in our patient considered a risk factor for neurodevelopmental disorders with a low penetrance (<10%) and variable expressivity [27,28,29]. These frequent genomic variations should be further explored to understand how their detection could be improved using nanopore sequencing. For these reasons, in many cases, the complementary use of orthogonal methods is required for validation.

Additionally, the cost of nanopore sequencing, while decreasing, can still be prohibitive for some large-scale projects.

In summary, whole-genome sequencing with long reads represents a valuable tool for improving diagnostic accuracy in rare diseases, especially in complex or inconclusive IRD cases. Its ability to detect structural variation, assemble haplotypes, and detect complex genomic rearrangements underscores its potential for molecular diagnostics and genetic counseling.

## 4. Materials and Methods

### 4.1. Subjects and Previous Studies

To test the clinical applications of this technique in IRD cases, we selected a cohort of patients who had previously undergone clinical exome sequencing (CES) using either Clinical Exome Solution (Sophia Genetics, Boston, MA, USA) or TruSight One (Illumina, San Diego, CA, USA), and/or whole-exome sequencing (WES), as described in Perea-Romero et al. [2]. Additionally, array comparative genomic hybridization (aCGH) was performed in one patient to validate a copy number variant (CNV) detected in CES.

From exome-sequencing previously studied patients with a suspicion of IRD from our cohort, we selected the following: (1) five IRD cases with previously identified variants used as a validation dataset; (2) one patient with a monoallelic pathogenic/likely pathogenic variant in recessive genes; (3) seven patients carrying variants of uncertain significance in recessive genes; (4) eight more non-informative cases, in which previous tests had not yielded conclusive results; (5) one patient with a low penetrance structural variant (SV); and, (6) one patient carrying both a pathogenic/likely pathogenic variant in a recessive gene and a structural variant (SV) of uncertain significance.

In total, 24 patients were selected for the study. Of these, 23 were from the Fundación Jiménez Díaz (FJD) University Hospital (Madrid, Spain), drawn from a cohort of 5372 families with inherited retinal dystrophy (IRD) that have been followed since 1990 and updated until February 2025 (as noted in Perea-Romero et al. [2]), while one patient was from the La Fe University Hospital (Valencia, Spain) IRD cohort. Informed consent was obtained from all participants, and the study adhered to the principles of the Declaration of Helsinki. Ethical approval was granted by both the FJD Research Ethics Committee (Approval No.: PIC172-20_FJD) and the Research Ethics Committee of La Fe University Hospital (Approval No.: 2022-276-1).

### 4.2. Clinical Evaluation

The clinical diagnosis was based on the patient’s medical history and ophthalmic examination. Specific questionnaires and electronic medical records were also reviewed. Patients were classified according to the area where the primary alterations in the retina occurred and whether extraocular symptoms were present, as previously described in Perea-Romero et al. [2].

### 4.3. Library Preparation and Genomic Sequencing

Whole-genome libraries were prepared using 1 μg of high-molecular-weight genomic DNA (HWM-DNA) from frozen blood cells using the EZ1 Advanced XL extraction system (QIAGEN) according to the manufacturer’s instructions. DNA integrity of >10 kb fragments was ensured via gel electrophoresis. The library was then prepared using the Ligation Sequencing Kit SQK-LSK109 (Oxford Nanopore Technologies, ONT; Oxford, UK), as described in Damián et al. [30].

### 4.4. Data Analyses

The data analysis prioritized the confirmation of previously reported variants within the cohort. Since all patients had already undergone short-read NGS studies that ruled out SNVs and CNVs in genes known to be associated with IRDs, the focus shifted to investigating SVs in long-read sequencing data. If a newly identified SV in an autosomal recessive gene was found to potentially explain the phenotype, additional SNV analysis was conducted at that locus.

Guppy was used to perform basecalling (Available online: https://github.com/nanoporetech/pyguppyclient (accessed on 3 January 2022)). The alignment of the generated fastq files with the GRCh38/hg38 human genome assembly was performed using minimap2 [31]. Single nucleotide variants (SNVs) and insertion–deletion variants (indels) were called using PEPPER (PEPPER-Margin-DeepVariant, [32]) and Clair3 [33]. SV calling was performed using the CuteSV [34], SVIM [35], Sniffles, and Sniffles2 [36,37], with the workflow reported in Damián et al. [30]. The in-house pipeline NextVariantFJD (Available online: https://github.com/TBLabFJD/NextVariantFJD, accessed on 17 April 2023) [38] was used for variant annotation, incorporating the variant effect predictor (VEP) [39] for point genetic variation (SNVs and indels) and AnnotSV [40] for SVs, along with additional custom information.

The American College of Medical Genetics and Genomics (ACMG) guidelines [41,42,43] were applied for variant prioritization. Previous disease associations were assessed in ClinVar (Available online: https://www.ncbi.nlm.nih.gov/clinvar/, accessed on 10 February 2025), the Human Gene Mutation Database (HGMD, Professional 2023.4; Available online: https://digitalinsights.qiagen.com/products-overview/clinical-insights-portfolio/human-gene-mutation-database/, accessed on 10 February 2025) and the Leiden Open Variation Database v.3.0 (LOVD; Available online: https://www.lovd.nl/, accessed on 10 February 2025). For variant filtering, we applied a minor allele frequency <0.01 in the following general population databases: GnomAD v4.1.0 (Genome Aggregation Database; Available online: http://gnomad.broadinstitute.org/, accessed on 10 February 2025), Kaviar (Known VARiants), and CSVS (Collaborative Spanish Variant Server) [44]. Structural variation was also filtered by frequency within the sequencing batch, retaining unique variants from a cohort of 34 samples.

Reported SVs were visually investigated using the Integrative Genomics Viewer (IGV) [45] and Ribbon [46] visualization tools. Coverage distribution analysis was performed using the mean coverage of 5 kb windows normalized to the mean coverage obtained for each patient and plotted against the rest of the cohort. The SV nomenclature was assigned according to ISCN 2020 guidelines [47].

Haplotype assembly was performed using the WhatsHap v2.3 software [48] with the recommended workflow on the bam files of a selected chromosome. The generated haplotagged bam files were then visualized in IGV tool.

### 4.5. Variant Validation

Sanger sequencing was used for candidate pathogenic variant confirmation, and segregation in relatives was performed whenever possible.

Copy number variations (CNVs) were confirmed via multiplex ligation-dependent probe amplification (MLPA; MRC-Holland, Amsterdam, the Netherlands) kits for the EYS gene or via comparative genomic hybridization array (aCGH) using the aCGX 60K platform (CGXTM SurePrint Technology, Agilent, Santa Clara, CA, USA) following the manufacturer’s protocol. The array images were scanned and extracted using the SureScan Microarray Scanner (Agilent Technologies, Santa Clara, CA, USA). CNV analysis was conducted using the Genoglyphix^®^ platform (Revvity, Inc, Boston, MA, USA). Karyotype analysis was conducted following the standard procedures to explore the detected SVs.

## 5. Conclusions

In conclusion, long-read whole-genome nanopore sequencing proved to be a valuable tool for characterizing two additional patients and offering a more detailed analysis of structural variation and haplotype reconstruction within the cohort. This approach enabled the identification of complex genomic rearrangements that were missed using conventional NGS. However, the failure to detect two previously reported variants highlights the current limitations of the technology and bioinformatics pipelines in capturing certain types of genomic variation, emphasizing the need for the continued optimization of sequencing protocols, data processing algorithms, and variant-calling tools. Despite these challenges, nanopore long-read sequencing remains an invaluable tool for resolving diverse genomic variations, particularly in clinical genetics, where precise variant determination is crucial for accurate molecular diagnosis. Moreover, its ability to provide faster and more precise molecular diagnoses has the potential to significantly impact patient care by facilitating tailored treatment options and improving health outcomes.

## Figures and Tables

**Figure 1 ijms-26-03825-f001:**
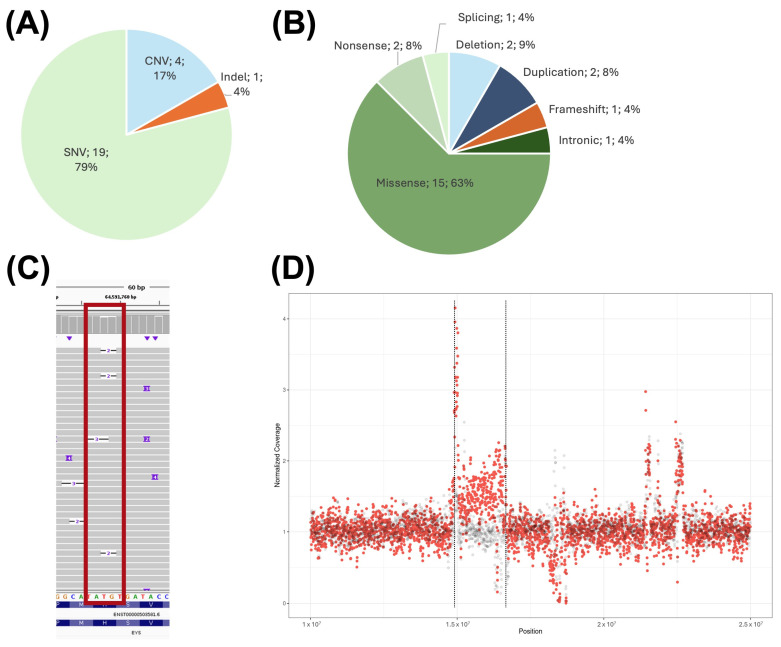
Distribution of previously reported variants by type (**A**) and mutation consequence (**B**). (**C**) IGV visualization of a previously reported indel in the C001VXB patient, genomic location is surrounded by a red rectangle. (**D**) Coverage distribution of LR-WGS data in the chr16:10000000–25000000 region containing the previously detected duplication of the C00231A patient (red) compared to the C00231G patient (gray), which was considered a negative control. Possible duplication boundaries are marked with a dotted line.

**Figure 2 ijms-26-03825-f002:**
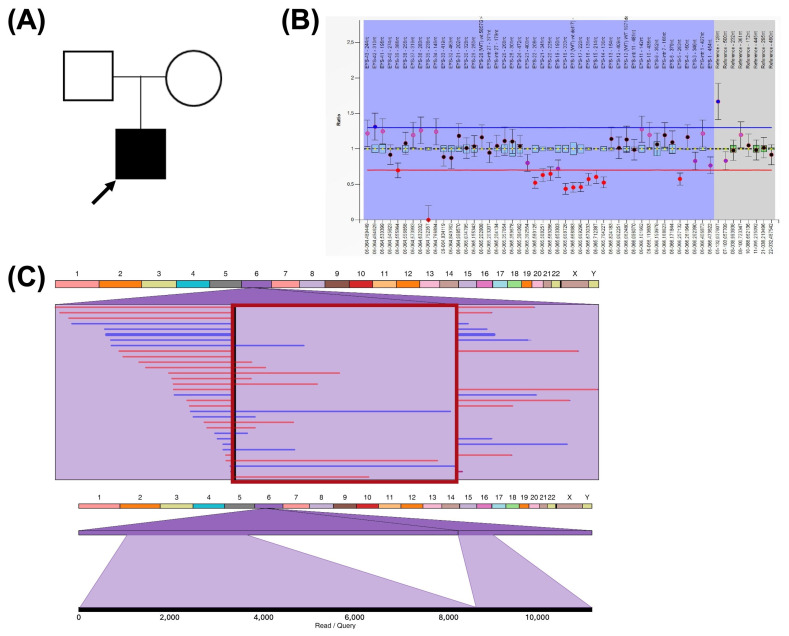
(**A**) Pedigree of C001VXB patient and his parents. Squares indicate males, circles indicate females, and the arrow denotes the proband. (**B**) Visualization of detected deletion in the *EYS* gene using MLPA analysis. (**C**) Ribbon visualization of the deletion in LR-WGS data. In the top image, the red rectangle frames both breakpoints with split reads on both sides. The bottom image shows the alignment of a selected single read that contains the deletion.

**Figure 3 ijms-26-03825-f003:**
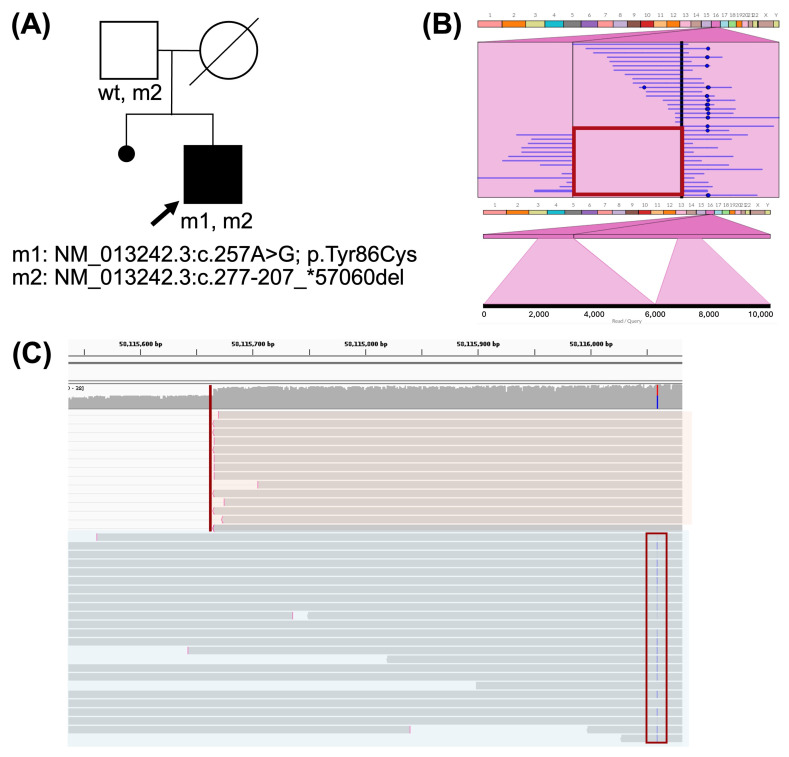
(**A**) Pedigree of C002312 patient and his parents. Squares indicate males, circles indicate females, and the arrow denotes the proband. Sanger segregation results are shown underneath each individual. (**B**) Ribbon visualization of the deletion. In the top image, the red rectangle frames both breakpoints, showing split reads on both sides. The bottom image displays the alignment of a selected single read that contains the deletion. (**C**) Visualization of sequences aligned with CFAP20 on IGV. Reads containing the SNV, framed on a red rectangle, are highlighted in light blue. Reads containing the deletion breakpoint, indicated by a red line, are highlighted in light orange.

**Figure 4 ijms-26-03825-f004:**
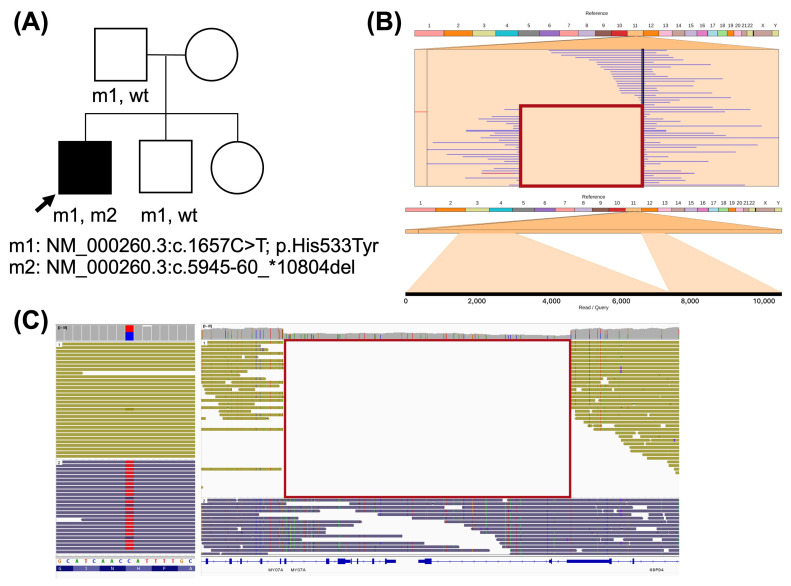
(**A**) Pedigree of C00231G patient and his parents. Squares indicate males, circles indicate females, and the arrow denotes the proband. Sanger segregation results are shown underneath each individual. (**B**) Ribbon visualization of the deletion. In the top image, the red rectangle frames both breakpoints, showing split reads on both sides. The bottom image displays the alignment of a selected single read that contains the deletion. (**C**) Visualization of sequences aligned with *MYO7A* on IGV. Reads are grouped and colored by haplotype. On the right image, the red rectangle frames both breakpoints of the deletion, showing split reads on both sides.

**Figure 5 ijms-26-03825-f005:**
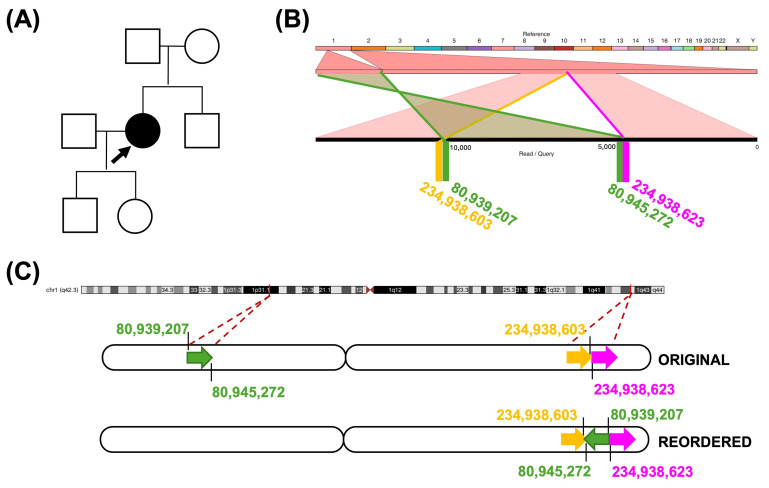
(**A**) Pedigree of the C001VWE patient. Squares indicate males, circles indicate females, and the arrow denotes the proband. (**B**) Ribbon visualization of a selected read containing the inverted insertion. (**C**) Schematic representation of chromosome 1 reorganization detected in the patient.

**Figure 6 ijms-26-03825-f006:**
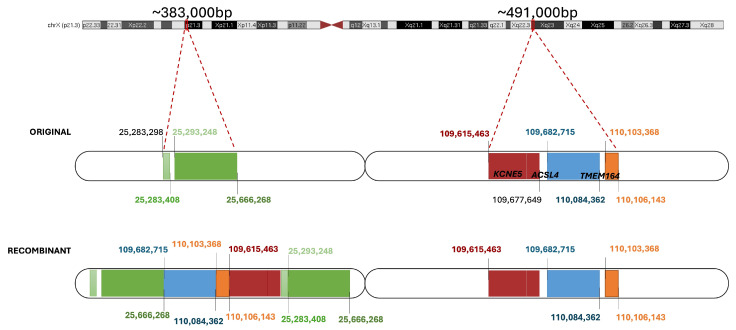
Schematic representation of the recombinant X chromosome rearrangement identified in the C001VWE patient. The region sizes are described above. Colored breakpoints were called, and black breakpoints were identified upon manual exploration.

## Data Availability

The genomic data are available on the European Genome-phenome Archive (EGA).

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
