# Peer review of "Long-Read Whole-Genome Sequencing as a Tool for Variant Detection in Inherited Retinal Dystrophies"

_ijms, 2025, doi:10.3390/ijms26083825_

Round 1
Reviewer 1 Report
Comments and Suggestions for Authors
This is an analysis of LR-WGS usage in patients with clinically diagnosed IRDs who previously had WES or CES testing. It could be very interesting but the results show that LR-WGS was only able to identify more precisely some (not all) of SVs and had no added diagnostic value.
And this is even not mentioned in the conlcusions.
The group of patients is very small, clinically heterogenous and with no clinical and pedigree data described for majority of them.
Here are my most important remarks:
- For majority of patients there are no clinical and pedigree data. There are also no information regarding stated or suspected clinical diagnosis. These data must be added in Table 1 or in a new table because without them it is not possible to analyse the
- Table 1 informs that the final causative diagnosis was done in 7 out of 23 patients (30%). This very low success rate raises doubts that the clinical diagnosis of IRD was incorrect. In my cohort of patients with clinically suspected IRDs the diagnostic efficiency after retinal NGS panel covering 367 genes is over 80% ! So the precise clinical data must be added (age at diagnosis, VA, visual field, nyctalopia/photophobia, OCT, ERG, clinical diagnosis etc.)
- I don’t know why some of monoallelic variants in BEST1, IMPG2 and PROM1 genes that are responsible for autosomal dominant disorders are not considered to be diagnostic. Don’t they fit the clinical and pedigree data or family analysis showed they are present in healthy relatives?
- Why is the variant in ARHGEF16 gene reported, if this gene is not related to any IRD?
- The conclusions must contain much more information (included some bad regarding limitations of the method):
- LR-WGS did not allow to identify any diagnostic variants previously not found in WES/CES so in this group of patients it had no clinically important added value.
- LR-WGS did not allow to identify variants previously found in WES/CES !!!
This is an analysis of LR-WGS usage in patients with clinically diagnosed IRDs who previously had WES or CES. It could be very interesting but the results show that LR-WGS was only able to identify more precisely some (not all) of SVs and had no added diagnostic value.
The group of patients is very small, clinically heterogenous and with no clinical and pedigree data described for majority of them.
Here are my important remarks:
- For majority of patients there are no clinical and pedigree data. There are also no information regarding stated or suspected clinical diagnosis. These data must be added in Table 1 or in a new table because without them it is not possible to analyse the
- Table 1 informs that the final causative diagnosis was done in 7 out of 23 patients (30%). This very low success rate raises doubts that the clinical diagnosis of IRD was incorrect. In my cohort of patients with clinically suspected IRDs the diagnostic efficiency after retinal NGS panel covering 367 genes is over 80% ! So the precise clinical data must be added (age at diagnosis, VA, visual field, nyctalopia/photophobia, OCT, ERG, clinical diagnosis etc.)
- I don’t know why some of monoallelic variants in BEST1, IMPG2 and PROM1 genes that are responsible for autosomal dominant disorders are not considered to be diagnostic. Don’t they fit the clinical and pedigree data or family analysis showed they are present in healthy relatives?
- Why is the variant in ARHGEF16 gene reported, if this gene is not related to any IRD?
- The conclusions must contain much more information (included some bad regarding limitations of the method):
- LR-WGS did not allow to identify any diagnostic variants previously not found in WES/CES so in this group of patients it had no clinically important added value.
- LR-WGS did not allow to identify variants previously found in WES/CES !!!
And please explain every abbreviation when it is first used in the text (some of them are present in the list of abbreviations only and some are not explained at all)
Author Response
This is an analysis of LR-WGS usage in patients with clinically diagnosed IRDs who previously had WES or CES testing.
It could be very interesting but the results show that LR-WGS was only able to identify more precisely some (not all) of SVs and had no added diagnostic value. And this is even not mentioned in the conlcusions.
The group of patients is very small, clinically heterogenous and with no clinical and pedigree data described for majority of them.
Here are my most important remarks:
Comment1: For majority of patients there are no clinical and pedigree data. There are also no information regarding stated or suspected clinical diagnosis. These data must be added in Table 1 or in a new table because without them it is not possible to analyse the
Response1: We thank the reviewer for this insightful and constructive comment.
The initial purpose of this paper was not to report a series of cases studied by WGS-long-reads, but just to show and comment the applications and limitations of this technique. However, your question raises to us the opportunity to publish the whole series adding all the clinical and genetic data.
Therefore, as suggested, we have now included a new supplementary table (Table S1) describing the extended phenotype and clinical data of all 24 patients included in our study. We are especially grateful for this suggestion, as it has significantly improved the clarity and completeness of the manuscript by providing a more comprehensive overview of the cohort and supporting a better interpretation of the results.
Comment2: Table 1 informs that the final causative diagnosis was done in 7 out of 23 patients (30%). This very low success rate raises doubts that the clinical diagnosis of IRD was incorrect. In my cohort of patients with clinically suspected IRDs the diagnostic efficiency after retinal NGS panel covering 367 genes is over 80% ! So the precise clinical data must be added (age at diagnosis, VA, visual field, nyctalopia/photophobia, OCT, ERG, clinical diagnosis etc.)
Response 2: We appreciate the reviewer’s comment and would like to clarify that the studied cohort consists of 5 control patients and 19 previously uncharacterized cases. These patients are not naïve for the genetic studies, as they were previously analyzed using short-read NGS approaches, such as clinical exome sequencing (CES) and/or whole exome sequencing (WES). The goal of the current study was to perform a more in-depth genomic analysis using long-read whole-genome sequencing (LR-WGS) to assess its potential in detecting structural variants that may have been missed by previous approaches.
As such, our diagnostic rate should not be directly compared with that of naïve cohorts. Rather, it aligns with similar studies using long-read sequencing in previously studied but unsolved cases. Related information about these studies has now been included in the manuscript, under the Discussion section, in lines 220-234: “Through this long-read whole-genome sequencing (LR-WGS) study, we were able to molecularly characterize two additional patients out of 19 previously unsolved index cases, representing a diagnostic yield of 10.5%. This result is comparable to that reported by Negi et al., who achieved an 11.2% diagnostic rate (11/98) in patients with suspected rare monogenic disorders that had remained unsolved after short-read NGS [15]. Similarly, another study focusing on monoallelic EYS cases previously analyzed by NGS identified a genetic diagnosis in 13.3% of the cohort (2/15) using long-read WGS [16].”
The clinical details of the cohort are provided in the new Table S1 for reference. Notably, our previous work (Perea-Romero et al., 2021. PMID: 33452396) demonstrated a diagnostic yield of approximately 70% using short-read NGS in IRD patients. The aim of this study is to complement that work by applying long-read sequencing to further resolve the remaining unsolved cases and explore its value in uncovering additional causative variants.
Comment3: I don’t know why some of monoallelic variants in BEST1, IMPG2 and PROM1 genes that are responsible for autosomal dominant disorders are not considered to be diagnostic. Don’t they fit the clinical and pedigree data or family analysis showed they are present in healthy relatives?
Response3: We sincerely thank the reviewer for this insightful comment, which prompted us to expand and clarify the information related to variant interpretation. In response, we have now included a more detailed explanation in the Results section (lines 103–114): “Exome sequencing in index patients C002319 and C002311 identified variants in ADGRV1 and PROM1, respectively. Segregation analysis revealed that the ADGRV1 variants were in cis, forming a complex allele in the patient, while the PROM1 variant was inherited from an unaffected mother, thereby excluding a dominant inheritance pattern. Additionally, two variants were found in other genes known to be associated with both autosomal dominant and recessive inheritance: BEST1 and IMPG2. However, the available information from genomic databases supported a recessive inheritance for both. The BEST1 variant affects the p.Pro101 residue, where other pathogenic variants (c.301C>A and c.302C>T) have been linked to autosomal recessive BEST1-related conditions (ClinVar IDs: 99710 and 877528). The IMPG2 variant (NM_016247.4:c.1300C>T) is observed in general population databases at a frequency (GnomAD v4, f = 1.2×10⁻⁴) higher than expected for a dominant pathogenic variant, supporting a non-dominant model of inheritance”.
We have also updated Table S2 to include the ACMG classification criteria applied to each variant, thereby strengthening the interpretation framework. Some index patients were found to carry variants that, upon reanalysis, were reclassified as variants of uncertain significance or likely benign. Nonetheless, we included all these cases as internal controls to help assess the performance of long-read whole-genome sequencing (LR-WGS) in detecting single nucleotide variants (SNVs). We are grateful to the reviewer for this valuable suggestion, which has significantly improved the quality of the manuscript.
Comment4: Why is the variant in ARHGEF16 gene reported, if this gene is not related to any IRD?
Response4: We appreciate the reviewer’s observation and fully agree with this point. Given the lack of established clinical association between ARHGEF16 and any known disease, the index case has now been reclassified as initially non-informative. This change has been reflected in the main text (line 137), which now reads: “CES analysis on the patient did not yield a conclusive molecular diagnosis, while WES identified a heterozygous truncating variant in the candidate gene ARHGEF16. Although this gene is not currently associated with any disease, its ubiquitous expression in human tissues initially suggested it as a potential contributor to the patient’s syndromic phenotype.”
Comment5: The conclusions must contain much more information (included some bad regarding limitations of the method):
- LR-WGS did not allow to identify any diagnostic variants previously not found in WES/CES so in this group of patients it had no clinically important added value.
- LR-WGS did not allow to identify variants previously found in WES/CES !!!
Response5:
We thank the reviewer for their suggestion. The conclusion has been revised to explicitly reflect both the strengths and limitations of our study, as follows: “In conclusion, long-read whole-genome nanopore sequencing proved to be a valuable tool for characterizing two additional patients and offering a more detailed analysis of structural variation and haplotype reconstruction within the cohort. This approach enabled the identification of complex genomic rearrangements that were missed using conventional NGS. However, the failure to detect two previously reported variants highlights the current limitations of the technology and bioinformatics pipelines in capturing certain types of genomic variation, emphasizing the need for continued optimization of sequencing protocols, data processing algorithms, and variant-calling tools. Despite these challenges, nanopore long-read sequencing remains an invaluable tool for resolving diverse genomic variations, particularly in clinical genetics, where precise variant determination is crucial for accurate molecular diagnosis. Moreover, its ability to provide faster and more precise molecular diagnoses has the potential to significantly impact patient care by facilitating tailored treatment options and improving health outcomes.”
Comment6: And please explain every abbreviation when it is first used in the text (some of them are present in the list of abbreviations only and some are not explained at all)
Response6: Abbreviations have been carefully reviewed and updated in the manuscript.
Reviewer 2 Report
Comments and Suggestions for Authors
This is a study of 23 patients with retinal dystrophy using long-read genome sequencing by nanopore technology. The overall approach is acceptable but the presentation of the methods and results are lacking details that are important. There is no explanation for the recruitment strategy, inclusion criteria, exclusion criteria, especially that there is access to a huge database? Did all 23 have exome sequencing and other testing? Is that the only criterion for inclusion? There is no mention how the previous results were obtained, which test etc? The refinement of a duplication is not impressive as a step up from the previous report and did not change any diagnosis. It is not clear if the CFAP20 case was missed because the testing was done earlier than the description in the literature or not. In most cases, details about previous testing approach and what the long-read genome sequencing added needs to be clearly outlined.
Author Response
This is a study of 23 patients with retinal dystrophy using long-read genome sequencing by nanopore technology. The overall approach is acceptable but the presentation of the methods and results are lacking details that are important.
Comment1: There is no explanation for the recruitment strategy, inclusion criteria, exclusion criteria, especially that there is access to a huge database? Did all 23 have exome sequencing and other testing? Is that the only criterion for inclusion?
Response1: To address the reviewer's concern, we have now explained in depth the recruitment strategy and inclusion/exclusion criteria followed in our study in the Materials and Methods section, between lines 288 and 310: “To test the clinical applications of this technique in IRD cases, we selected a cohort of patients who had previously undergone clinical exome sequencing (CES) using either Clinical Exome Solution (Sophia Genetics, USA) or TruSight One (Illumina, USA), and/or whole-exome sequencing (WES), as described in Perea-Romero et al. [2]. Additionally, array comparative genomic hybridization (aCGH) was performed in one patient to validate a copy number variant (CNV) detected in CES.
From exome-sequencing previously studied patients with suspicion of IRD from our cohort, we selected: 1) five IRD cases with previously identified variants were used as a validation dataset; 2) one patient with a monoallelic pathogenic/likely pathogenic variant in a recessive genes; 3) seven patients carrying variants of uncertain significance in recessive genes; 4) eight more non-informative cases, in which previous tests had not yielded conclusive results; 5) one patient with a low penetrance structural variant (SV) was included; and, 6) one patient carrying both a pathogenic/likely pathogenic variant in a recessive gene and a structural variant (SV) of uncertain significance.
In total, 24 patients were selected for the study. Of these, 23 were from the Fundación Jiménez Díaz (FJD) University Hospital (Madrid, Spain), drawn from a cohort of 5,372 families with inherited retinal dystrophy (IRD) that has been followed since 1990 and updated until February 2025 (as noted in Perea-Romero et al. [2]), while one patient was from the La Fe University Hospital (Valencia, Spain) IRD cohort. Informed consent was obtained from all participants, and the study adhered to the principles of the Declaration of Helsinki. Ethical approval was granted by both the FJD Research Ethics Committee (Approval No.: PIC172-20_FJD) and the Research Ethics Committee of La Fe University Hospital (Approval No.: 2022-276-1).”
Comment2: There is no mention how the previous results were obtained, which test etc?
Response2: We want to thank the reviewer for their insightful comment. To address their concern, while all included patients had previously undergone clinical exome sequencing and/or whole-genome sequencing, we have added a new supplementary table (Table S1) to the manuscript. This table provides a detailed of the previous molecular studies performed for each patient, which we hope will clarify any uncertainties.
Comment3: The refinement of a duplication is not impressive as a step up from the previous report and did not change any diagnosis.
Response3: We want to thank the reviewer’s thoughtful comment, and we appreciate the opportunity to clarify this point. In our study, we report two previously identified deletions and two duplications within our cohort. We described in more detail a duplication in C00231A patient, that could not be precisely defined with LR-WGS. Additionally, we also explained one deletion involving the EYS gene in patient C001VXB, who had already been molecularly characterized. The inclusion of validation cases was specifically aimed for evaluating the performance of the long-read sequencing technology and bioinformatic pipeline in variant detection.
Although this deletion had been previously diagnosed, long-read sequencing provided more precise breakpoint identification, offering deeper insights into the potential impact on the resulting protein. In contrast, CNV detection using short-read sequencing typically relies on coverage comparisons between samples, which can be imprecise and ambiguous when determining the exact size of a CNV. Long-read sequencing, however, allows us to directly identify the structural variation and define breakpoints without the need for sample comparisons. While this refinement did not alter the molecular diagnosis, it contributed valuable information about the variant's precise genomic context and possible protein effect. Furthermore, the detailed breakpoint analysis could assist in segregation studies within affected families. We believe these findings underscore the utility of long-read sequencing in providing more accurate and clinically relevant data.
To follow the reviewer’s recommendation that this finding is not impressive, we have included the reference to C001VXB identified deletion within the “Confirmation of previously detected variants in the studied cohort” subsection (line 89) of Results.
Comment4: It is not clear if the CFAP20 case was missed because the testing was done earlier than the description in the literature or not.
Response4: The reviewer makes a good point. We have addressed this in the manuscript to clarify the timing of the gene’s association, specifically in line 140: “After LR-WGS, two variants were identified in a recently associated IRD gene (May 2024 PanelApp), previous NGS studies were performed before 2018.” This timing discrepancy highlights the evolving nature of genetic research, where previously unrecognized genes are identified as being linked to specific conditions, which underscores the importance of continually updating genetic databases and diagnostic approaches. We hope this explanation provides additional context regarding the findings and their relevance to the patient's diagnosis.
Comment5: In most cases, details about previous testing approach and what the long-read genome sequencing added needs to be clearly outlined.
Response5: We sincerely thank the reviewer for their insightful comment. We intend to provide more clarity regarding the previous testing approaches in lines 290-295 of Materials and Methods: “To test the clinical applications of this technique in IRD cases, we selected a cohort of patients who had previously undergone clinical exome sequencing (CES) using either Clinical Exome Solution (Sophia Genetics, USA) or TruSight One (Illumina, USA), and/or whole-exome sequencing (WES), as described in Perea-Romero et al. [2]. Additionally, array comparative genomic hybridization (aCGH) was performed in one patient to validate a copy number variant (CNV) detected in CES” and Table S2. Additionally, long-read genome sequencing (LR-WGS) in this study provided confirmation of both single-nucleotide variants and structural variants previously identified in the study cohort. Additionally, precise breakpoint identification for structural variants, which could not be fully resolved by short-read sequencing methods. Moreover, LR-WGS allowed for genotype definition in recessive inherited retinal dystrophy (IRD) patients, detection of mobile element translocations and structural rearrangement identification. We have provided a summary of what LR-WGS provides for each specific patient in Table S2. We believe these additions strengthen the manuscript by emphasizing the unique contributions of LR-WGS to variant detection and genomic analysis.
Round 2
Reviewer 2 Report
Comments and Suggestions for Authors
none